

# The SIR dynamic model of infectious disease transmission and its analogy with chemical kinetics

Cory M. Simon

School of Chemical, Biological, and Environmental Engineering, Oregon State University, Corvallis, OR, USA

## ABSTRACT

Mathematical models of the dynamics of infectious disease transmission are used to forecast epidemics and assess mitigation strategies. In this article, we highlight the analogy between the dynamics of disease transmission and chemical reaction kinetics while providing an exposition on the classic Susceptible–Infectious–Removed (SIR) epidemic model. Particularly, the SIR model resembles a dynamic model of a batch reactor carrying out an autocatalytic reaction with catalyst deactivation. This analogy between disease transmission and chemical reaction enables the exchange of ideas between epidemic and chemical kinetic modeling communities.

## INTRODUCTION

Mathematical models of the dynamics of infectious disease transmission (*Brauer, 2017*; *Hethcote, 2000*) are useful for forecasting epidemics, evaluating public health interventions, and inferring properties of diseases.

In compartmental epidemic models (*Brauer, 2008*), each member of the population is categorized based on their disease status in addition to, possibly, their attributes and/or the treatment they received. The dynamics of disease transmission are then typically modeled with differential equations that describe the flow of individuals between the compartments as the population mixes, the disease spreads, infected individuals progress through the stages of the disease, and public health interventions are implemented.

The parsimonious, classic Susceptible–Infectious–Removed (SIR) compartmental model (*Kermack & McKendrick, 1927*; *Anderson, 1991*) gives insights into the dynamics of epidemics and shows utility for understanding how public health interventions affect the trajectory of an epidemic (*Bertozzi et al., 2020*).

In this article, we highlight the analogy between the dynamics of disease transmission and chemical reaction kinetics while formulating and analyzing the SIR epidemic model. This article is pedagogical in nature and its aim, in making the connection between chemical kinetics and the spread of infectious disease, is knowledge exchange across the two disciplines of chemical kinetic and epidemic modeling.

Corresponding author
Cory M. Simon,
cory.simon@oregonstate.edu

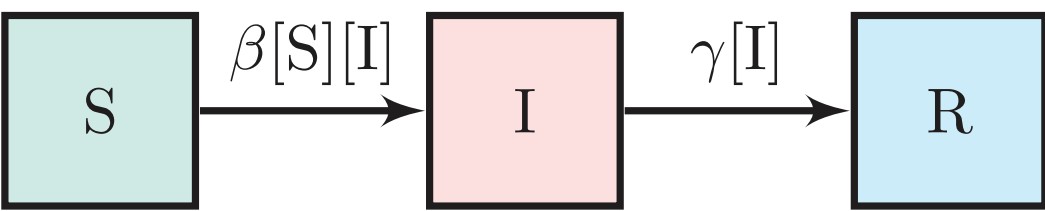

**Figure 1** **The SIR model.** The boxes represent the set of susceptible, infectious, and removed individuals. The arrows represent flow and are annotated with per capita flow rates.

## RESULTS

### The SIR model

In the classic SIR model of an epidemic (*Kermack & McKendrick, 1927*; *Murray, 1993*; *Hethcote, 2000*, *Edelstein-Keshet, 1988*; *Frauenthal, 2012*; *Brauer, Castillo-Chavez & Feng, 2019*), each member of the population belongs to one of three compartments: Susceptible, Infectious, or Removed.

#### *The reactions*

Susceptible folks can contract the disease if they come into contact with an infectious individual. Once infected, they move into the infectious compartment, assuming zero delay between infection and the ability to transmit the disease. This is analogous to an irreversible, autocatalytic chemical reaction (*Schuster, 2019*; *Scott, 1983*) between a reactant, S, and catalyst, I:

$$S + I \rightarrow 2I \qquad \{1\}$$

Infectious individuals eventually recover or die from the disease, entering the removed compartment. Folks in the removed compartment do not participate in disease transmission. That is, they cannot transmit the disease, nor can they contract it again, assuming that recovery from the disease confers immunity to reinfection. This is analogous to a reaction where the catalyst, I, irreversibly degrades or converts to a deactivated form, R:

$$I \rightarrow R \qquad \{2\}$$

We assume *permanent* protective immunity is conferred upon recovery from the disease, thus neglecting the possibility of an R → S reaction.

So, the SIR model of an epidemic is analogous to an autocatalytic reaction (rxn. {1}) with catalyst deactivation (rxn. {2}). An infectious individual (the catalyst, I), (i) upon contacting (colliding with) a susceptible individual (the reactant, S), can convert them into another infectious individual (another catalyst particle) and (ii) recovers or dies (deactivates) with time. Figure 1 depicts the flow of individuals between compartments under the SIR model, induced by rxns. {1} and {2} .

#### *The dynamic mathematical model*

Mathematically, the SIR model (*Kermack & McKendrick, 1927*; *Edelstein-Keshet, 1988*; *Murray, 1993*; *Martcheva, 2015a*) is equivalent to a dynamic model of a well-mixed, isothermal batch reactor carrying out the two homogeneous, elementary rxns. {1} and {2} .

As in a (closed) batch reactor, we neglect immigration and emigration (hence, the absence of flow to/from external populations in Fig. 1). Moreover, we take births and deaths not caused by the disease to be negligible over the (short) time scale of the epidemic.

Let $[S](t)$, $[I](t)$ and $[R](t)$ be the fraction of the population that is susceptible, infectious, and removed, respectively, at time $t$. Considering a large population, we treat $[S]$, $[I]$, and $[R]$ as continuous variables. So, $[S]$, $[I]$, $[R] \in [0, 1]$.

*The incidence rate*

Assuming their spatial mixing is uniform (*Wilson & Worcester, 1945*; *Weiss, 2013*), we invoke the law of mass action to model the rate at which susceptible and infectious individuals "react" via bimolecular, autocatalytic rxn. {1}. The incidence rate of the disease, that is, the number of new infections per unit time (*Martcheva, 2015a*), is then $\beta [S][I]$ (per capita). A symmetric, bilinear function of $[S]$ and $[I]$, intuitively, the incidence rate doubles if $[I]$ ($[S]$) doubles while $[S]$ ($[I]$) is fixed. The second-order transmission rate constant $\beta > 0$ is the product of the average frequency of contacts of an individual in the population and the transmissibility of the disease (the probability of transmission conditioned upon contact).

*The recovery rate*

We model the rate at which infectious individuals "deactivate" (are removed) via rxn. {2} with first-order kinetics, that is, as $\gamma [I]$ (per capita). The inverse of the first-order recovery rate constant $\gamma > 0$ is the average time period that an infected individual is infectious (*Keeling & Rohani, 2011*).

The assumptions above are summarized by the rates of flow between the compartments shown in Fig. 1. We arrive at the set of nonlinear, coupled differential equations that comprise the SIR dynamic model of infectious disease transmission by writing a mass balance on each compartment in Fig. 1:

$$\frac{d[S]}{dt} = -\beta [S][I] \tag{1}$$

$$\frac{d[I]}{dt} = \beta [S][I] - \gamma [I] \tag{2}$$

$$\frac{d[R]}{dt} = \gamma [I]. \tag{3}$$

The only two parameters in the SIR model are the transmission and recovery rate constants, $\beta$ and $\gamma$, respectively. The average duration of infectiousness, $\gamma^{-1}$, could be estimated from contact tracing or shedding studies (*Fine, 2003*). The transmission rate constant, $\beta$, could be identified by fitting differential Eqs. (1)–(3) to epidemic time series data (case counts) (*Martcheva, 2015a*; *Weiss, 2013*), much like identifying a reaction rate constant from concentration time series (*Fogler, 2010*).

Addition of Eqs. (1)–(3) confirms the population is closed and demography is neglected, that is, $[S](t) + [I](t) + [R](t) = 1, \forall t \geq 0$. As a consequence, Eqs. (1) and (2) fully determine the SIR model dynamics, with $[R](t) = 1 - [S](t) - [I](t)$.

### The replacement and basic reproduction numbers

Two important numbers aid our characterization and understanding of SIR model dynamics: the replacement number, $r$, and the basic reproduction number, $\mathcal{R}_0$.

The *replacement number*, $r = r(t)$, is the expected number of folks (directly) infected by a typical infectious individual, mixing in the population, over the course of their infectiousness (*Hethcote, 2000*). Because the concentration of susceptible folks $[S] = [S](t)$ influences the frequency that a typical infectious individual contacts a susceptible individual, $r$ changes over the course of an epidemic. In the SIR model, a typical infectious individual is expected to produce $\beta [S](t)$ new infections per unit time (incidence rate per infectious individual) for an infectious duration of $\gamma^{-1}$. The replacement number is therefore:

$$r = r(t) = \frac{\beta}{\gamma} [S](t). \tag{4}$$

The *basic reproduction number*, $\mathcal{R}_0$, is defined as the initial replacement number when *one* infectious individual is introduced into an *all-susceptible* population (*Edelstein-Keshet, 1988*; *Hethcote, 2000*). In this context, $\mathcal{R}_0$ in the SIR model is the replacement number in Eq. (4) when $[S] \approx 1$:

$$\mathcal{R}_0 = \frac{\beta}{\gamma}. \tag{5}$$

That is, $\mathcal{R}_0$ is the expected number of infections directly caused by a single infectious individual introduced into an entirely susceptible population (*Hethcote, 2000*).

The dimensionless numbers $r$ and $\mathcal{R}_0$ are properties of both the disease and the population (*Delamater et al., 2019*). While $r = r(t)$ changes with time, $\mathcal{R}_0$ is constant and defined only at the initial stage of a particular context: when one infectious individual is introduced to an all-susceptible population. Notably, the two numbers are related via $r(t) = \mathcal{R}_0[S](t)$.

If the basic reproduction number $\mathcal{R}_0$ in Eq. (5) is large (small), the infected are infectious for a long (short) period of time, the disease is (not) easily transmitted, and/or the average frequency of contacts in the population is high (low). Under the analogy with chemical kinetics, since the activity and longevity of the catalyst, I, are embedded in $\beta$ and $\gamma$, respectively: $\mathcal{R}_0$ is large (small) if the catalyst has a high (low) activity and/or remains active for a long (short) time. These remarks also hold for the replacement number, $r = \mathcal{R}_0[S]$. However, $r$ decreases as the concentration of the reactant, $[S]$, decreases, owing to the reduced frequency that any given catalyst particle encounters a reactant particle to catalyze its conversion into another catalyst particle by rxn. {1} .

## SIR model dynamics

In the SIR model, what happens if we introduce a small number of infectious individuals into a large population of susceptible individuals? This is akin to injecting our deactivating autocatalyst, I, into a well-mixed batch reactor containing pure S. The corresponding initial conditions are:

$$[S](0) = [S]_0 \tag{6}$$

$$[I](0) = [I]_0 \tag{7}$$

$$[R](0) = 0, \tag{8}$$

with $[S]_0 + [I]_0 = 1$, $[I]_0 \ll 1$ and $[S]_0$, $[I]_0 > 0$. We consider $[R](0) = 0$ for the interesting case where a population is exposed to a novel pathogen to which it has no immunity.

### The replacement number r(t) determines the sign of [I]′(t)

The replacement number $r(t)$ in Eq. (4) is key to understanding SIR model dynamics. By inspection of Eq. (2), $[I](t)$ is increasing at time $t$ if the replacement number $r(t) > 1$ and decreasing if $r(t) < 1$. This is intuitive: if the typical infectious person mixing in the population is expected to infect less than one susceptible person before they recover, they are not expected to replace themselves with a new infectious individual to propagate the disease, and, consequently, $[I](t)$ is decreasing. Note $r(t) < 1 \Leftrightarrow [S](t) < \mathcal{R}_0^{-1}$.

### Does an epidemic ensue?

We first address a qualitative question: given the initial conditions in Eqs. (6)–(8), does the disease invade the population? The outcome depends on the initial replacement number $r_0 := r(0) = \mathcal{R}_0[S]_0$ in a threshold manner. If $r_0 > 1$, $[I](t)$ initially increases. That is, the disease spreads; an epidemic ensues. If $r_0 < 1$, $[I](t)$ initially decreases and, since $[S](t)$ is a decreasing function of time (see Eq. (1)), decays to zero. That is, the disease dies out; an epidemic does not ensue.

Under the chemical reaction analogy, if $r_0 < 1$ ($r_0 > 1$), the injected catalyst particles deactivate via rxn. {2} faster (slower) than they catalyze rxn. {1} to convert the reactant, S, into more catalyst, I, to propagate autocatalytic rxn. {1}; the reaction dies out (proceeds).

For our remaining analysis, we take $r_0 < 1$ and further analyze the dynamics of an SIR epidemic.

### Initial exponential growth

Early in the epidemic, the number of infectious folks grows approximately exponentially with growth rate $(r_0 - 1)\gamma$:

$$[I](t) \approx [I]_0 e^{(r_0-1)\gamma t}. \tag{9}$$

This follows from Eq. (2) if we neglect the depletion of the susceptible pool by approximating $[S](t) \approx [S]_0$, valid only in the initial stage of the epidemic; as the disease spreads, $[S]$ decreases and diminishes the replacement number. Equation (9) is thus an overestimate.

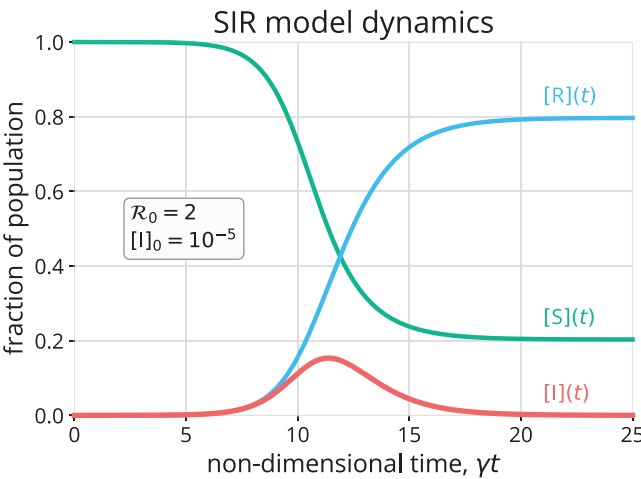

**Figure 2 Numerical approximation (*Rackauckas & Nie, 2017*) of the solution to the SIR model in Eqs. (1)–(3) with initial conditions in Eqs. (6)–(8).**

Since Eq. (9) is also a valid initial approximation for $r_0 < 1$, it reinforces that an epidemic will not ensue if $r_0 < 1$, since $[I](t)$ would then decay approximately exponentially.

### A simulation of an SIR model epidemic

Figure 2 shows the numerical solution to Eqs. (1)–(3) with initial conditions in Eqs. (6)–(8) for $\mathcal{R}_0 = 2$ and $[I]_0 = 10^{-5}$. Initially, the concentration of infectious folks, $[I]$, grows approximately exponentially (see Fig. S1 for a comparison of $[I](t)$ with Eq. (9)). The concentration of susceptible folks, $[S]$, decreases monotonically as the disease invades the population. As a result, the frequency with which any given infectious individual comes into contact with a susceptible individual decreases. In conjunction with the infectious folks recovering, this eventually causes the (net) growth rate of $[I]$ to diminish and, when the replacement number $r$ in Eq. (4) drops below one, causes $[I]$ to decay. The epidemic self-extinguishes, that is, $\lim_{t\to\infty}[I](t) = 0$ (*Bjørnstad et al., 2020a*). The R category accumulates the folks that have been infected by and have recovered or died from the disease, $[R](t) = \gamma \int_0^t [I](\tau)d\tau$ (see Eq. (3) with initial condition 8). Notably, the disease does not infect the entire population, even after an infinite amount of time ($\lim_{t\to\infty}[S](t) \neq 0$). That is, the epidemic self-extinguishes not because the population is depleted of susceptible folks, but rather because it is depleted of infectious folks (*Murray, 1993*; *Weiss, 2013*). An alternative visualization of SIR model dynamics in Fig. S2 emphasizes $[S](t) + [I](t) + [R](t) = 1, \forall t \geq 0$.

Under the chemical kinetics analogy, the autocatalytic rxn. {1} begins to slow down after the concentration of the reactant, $[S]$, decreases sufficiently. Once the replacement number $r(t)$ drops below one (i.e., once $[S]$ drops below $\mathcal{R}_0^{-1}$), the catalyst, I, is deactivating via rxn. {2} faster than it is converting the reactant, S, into more catalyst to replenish itself via rxn. {1}. Consequently, the catalyst concentration, $[I]$, begins to drop and decays to zero. Owing to catalyst deactivation (rxn. {2}), not all reactant, S, is consumed, even after an infinite amount of time.
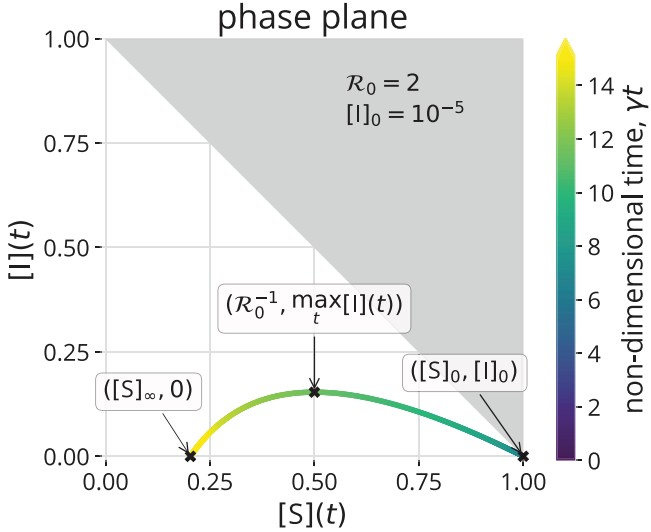

**Figure 3 The trajectory of the solution to the SIR model, with initial conditions in Eqs. (6)–(8), in the** ([S], [I]) **phase plane, given by Eq. (10).** The trajectory is colored according to time. The initial conditions, peak prevalence of infectious folks, and final conditions are marked. Since $[S](t) + [I](t) \leq 1$, points in the gray region are not feasible.

### Solution trajectory in the ([S], [I]) phase space

We can analytically find the trajectory of the solution to Eqs. (1) and (2) in the ([S], [I]) phase plane. Dividing Eq. (2) by Eq. (1) takes us into the phase plane by giving a differential equation with a $\frac{d[I]}{d[S]}$ derivative, with the view of [I] as a function of [S]. Separating, integrating, and applying the initial conditions in Eqs. (6) and (7), we arrive at the solution path (*Brauer, 2008*):

$$[I](t) = 1 - [S](t) + \frac{1}{\mathcal{R}_0} \log\left(\frac{[S](t)}{[S]_0}\right). \tag{10}$$

Figure 3 shows the trajectory of the solution given by Eq. (10) ($\mathcal{R}_0 = 2$, $[I]_0 = 10^{-5}$), traveling from the initial condition in the bottom right corner to the final condition on the bottom left. The trajectory reinforces that $[S](t)$ decreases monotonically with time; that $[I](t)$ increases, peaks, then diminishes to zero; and that a fraction of the population remains susceptible after the epidemic dies out.

### Final size of the epidemic

Given the epidemic (autocatalytic reaction) dies out before all of the susceptible folks (reactant) have been infected, what fraction of the population (reactant) remains susceptible (unreacted) after the epidemic ends?

We find an implicit equation for $[S]_\infty := \lim_{t\to\infty}[S](t)$ by taking the limit $t \to \infty$ in Eq. (10):

$$0 = 1 - [S]_\infty + \frac{1}{\mathcal{R}_0} \log\left(\frac{[S]_\infty}{[S]_0}\right). \tag{11}$$
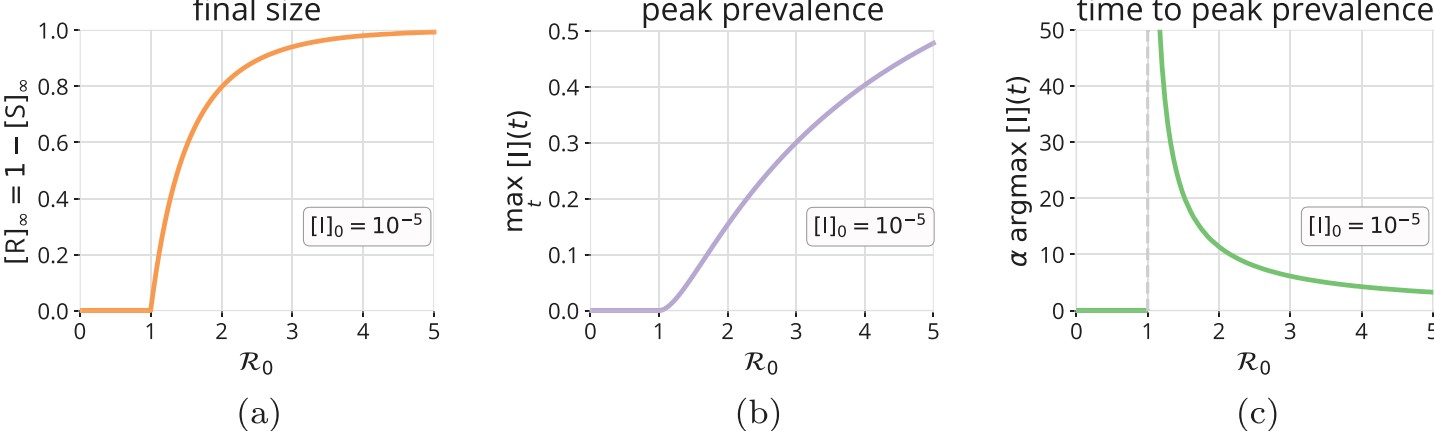

**Figure 4 Final size and peak prevalence of an SIR epidemic.** As a function of $\mathcal{R}_0$, (A) the fraction of the population that will be infected over the course of the epidemic, $[R]_\infty = 1 - [S]_\infty$, with $[S]_\infty$ obtained from Eq. (11), (B) the fraction of the population that is infectious at peak prevalence, $\max_t [I](t)$, via Eq. (12), and (C) the (non-dimensional) time to reach peak prevalence, $\text{argmax}_t \, \alpha[I](t)$, computed by a grid search using the numerical solution to the SIR model. All correspond to initial conditions in Eqs. (6)–(8).

We used the fact that the epidemic eventually dies out, $\lim_{t\to\infty}[I](t) = 0$. $[S]_\infty$ is the unique root of Eq. (11) in $(0, \mathcal{R}_0^{-1})$ (Hethcote, 2000). The fraction of the population infected over the course of the epidemic is $\lim_{t\to\infty}[R](t) =: [R]_\infty = 1 - [S]_\infty$ since the R category accumulates those that have recovered or died from the disease. Figure 4A shows that, as $\mathcal{R}_0$ increases from one, more of the population will be infected over the course of the epidemic.

### Peak prevalence of infectious folks

The peak prevalence of infectious folks, $\max_t [I](t)$, is important because it determines the maximum strain on the healthcare system. Both Eqs. (10) and (2) show that the maximum in $[I](t)$ (look for $\frac{d[I]}{d[S]} = 0$ or $\frac{d[I]}{dt} = 0$, respectively) occurs when the replacement number $r(t) = 1$. Before the peak is reached, the replacement number $r(t) > 1$ and $[I](t)$ is increasing; after the peak, $r(t) < 1$ and $[I](t)$ is decreasing to zero. Substituting $[S] = \mathcal{R}_0^{-1}$ into Eq. (10), we find:

$$\max_t[I](t) = 1 - \frac{1}{\mathcal{R}_0}\left[1 + \log\left(\mathcal{R}_0[S]_0\right)\right]. \tag{12}$$

Figure 4B shows how the peak prevalence of infectious folks increases as $\mathcal{R}_0$ increases from one. In addition, Fig. 4C shows that the (non-dimensional) time to reach peak prevalence, $\text{argmax}_t \, \alpha[I](t)$, decreases as $\mathcal{R}_0$ increases from one.

### Herd immunity

A population achieves *herd immunity* when a sufficient fraction are immune to the disease so as to confer indirect, population-level protection from an invasion of the disease upon the introduction of an infectious individual. Notably, the immunity could be acquired by either previous infection or by vaccination (Hethcote, 2000).

The administration of a vaccine that confers perfect and permanent immunity to a susceptible individual is modeled by introducing an S → R reaction to the SIR model, that
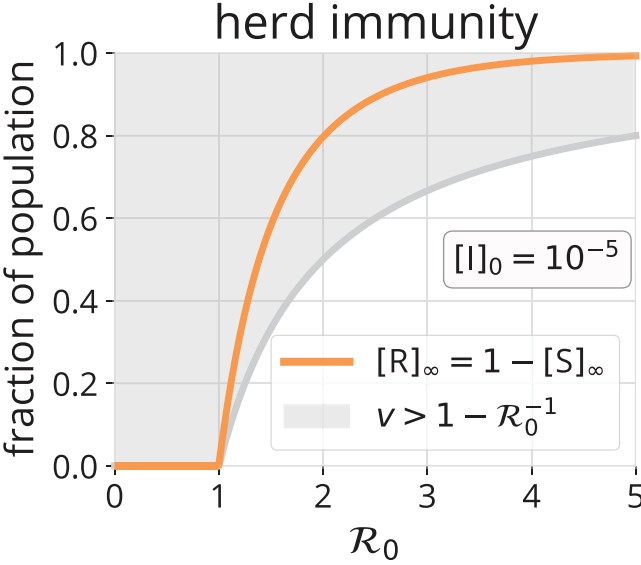

**Figure 5 Herd immunity.** The gray region shows the fraction $v > 1 - \mathcal{R}_0^{-1}$ of the population that must be immune to achieve herd immunity, as a function of $\mathcal{R}_0$. Such immunity could be conferred by vaccination and/or by previous infection. For comparison, the orange line shows $[R]_\infty$, the number of folks that will be infected over the course of an SIR epidemic without any vaccination.

is, by allowing flow in Fig. 1 from the susceptible category directly to the removed category (*Yusuf & Benyah, 2012*; *Martcheva, 2015a*; *Shulgin, Stone & Agur, 1998*).

What fraction $v$ of a population must be immune to the disease in order to achieve herd immunity? $[I](t)$ will decrease upon introducing an infectious individual if the replacement number $r(t) < 1$, that is, if $[S] < \mathcal{R}_0^{-1}$. Thus, a fraction $v > 1 - \mathcal{R}_0^{-1}$ of the population must be immune to achieve herd immunity. Figure 5 shows the region $v > 1 - \mathcal{R}_0^{-1}$ (gray) and illustrates that, as $\mathcal{R}_0$ increases from one, more of the population must be immune to prevent an epidemic via herd immunity.

In the chemical kinetics analogy, herd immunity results from reducing the concentration of the reactant, $[S]$, so that a catalyst particle, I, fed to the reactor is expected to deactivate before it encounters an S particle and autocatalyzes rxn. {1} to replace itself. That is, to achieve herd immunity, $[S]$ must be reduced sufficiently to make the replacement number $r$ less than one.

We now make a comparison between herd immunity achieved exclusively (i) by vaccination (S → R) and (ii) by infection and removal (recovery or death) (S → I → R). Consider an all-susceptible population. Under strategy (i), vaccinating a fraction $v = 1 - \mathcal{R}_0^{-1}$ of the population will suffice to confer herd immunity. Under strategy (ii), by introducing infectious individuals and allowing the epidemic to run its course, a fraction $[R]_\infty$ of the population will be infected over the course of the epidemic. Though, herd immunity will be achieved when $[S] = \mathcal{R}_0^{-1}$. At this point, $[I](t)$ is maximal (see Eq. (10) and Fig. 3). These infectious individuals will infect more susceptible folks before they all recover/die, even though, on *average*, each is expected to recover/die before they infect a susceptible individual ($r < 1$ after this point). The comparison of $v = 1 - \mathcal{R}_0^{-1}$ and $[R]_\infty$ in
Fig. 5 confirms that a significantly greater fraction of the population will be infected by the disease if we let it run its course than would have needed vaccination to achieve herd immunity. However, the most obvious benefit of the path S → R over S → I → R is avoidance of disease-induced suffering and, possibly, death.

### Conclusion: $\mathcal{R}_0$ is a useful tool

The dimensionless basic reproduction number $\mathcal{R}_0$ is a property of an infectious disease within a population. The average frequency of contacts of an individual in the population, the transmissibility of the disease, and the average duration of infectiousness are all embedded in $\mathcal{R}_0$. In the SIR model, $\mathcal{R}_0$ influences whether or not the disease invades the population, the initial exponential growth rate of infectious folks, how many are infected over the course of the epidemic, the peak prevalence of infectious folks, and how many must be vaccinated to achieve herd immunity.

## Extensions to the SIR model

The SIR model is a very simple epidemic model, but we can extend it to model other epidemiological factors and prevention/control measures by introducing:

### Births, deaths and loss of immunity

To model infectious disease transmission over longer time scales, we can modify the SIR model to account for births and deaths not caused by the disease by allowing flow into the S compartment and flow out of all compartments, respectively. See Fig. 6A (top). While Eqs. (1)–(3) resemble a dynamic model of a closed batch reactor carrying out rxns. {1} and {2}, the modified SIR model with demographics resembles a dynamic model of a continuously stirred tank reactor (Fogler, 2010): births are represented by a feed stream of pure S flowing into the reactor, while deaths are represented by an effluent stream. In addition, as opposed to assuming recovery from infection confers *permanent* immunity, we can model *temporary* immunity by introducing an R → S reaction to represent loss of immunity. Because births and the loss of immunity continuously add to the susceptible pool, [I](t) in the SIR model with births, deaths, and loss of immunity can exhibit damped oscillations that settle on an endemic equilibrium, where the disease remains in the population indefinitely (i.e., [I] is maintained at a non-zero value). See Fig. 6A (bottom). At an endemic equilibrium, the replacement number is maintained at one so that [I](t) is neither increasing nor decreasing (Hethcote, 2000; Bjørnstad et al., 2020b; Earn, 2008).

### Time-varying parameters

A time-varying transmission rate constant $\beta = \beta(t)$ can model (voluntary or policy-induced) changes in behavior during an epidemic that affect (i) the average frequency of contacts of an individual in the population or (ii) the transmissibility of the disease. Examples include staying home, social distancing, and taking hygiene measures (Fenichel et al., 2011; Reluga, 2010; Bjørnstad et al., 2020a; Bootsma & Ferguson, 2007). A periodic $\beta(t)$ can model seasonality of an infectious disease (Fisman, 2007; Aron & Schwartz, 1984). A time-varying recovery rate constant $\gamma = \gamma(t)$ can model changes in the average time period of infectiousness, for example, a reduction of $\gamma^{-1}$ by administering a drug to

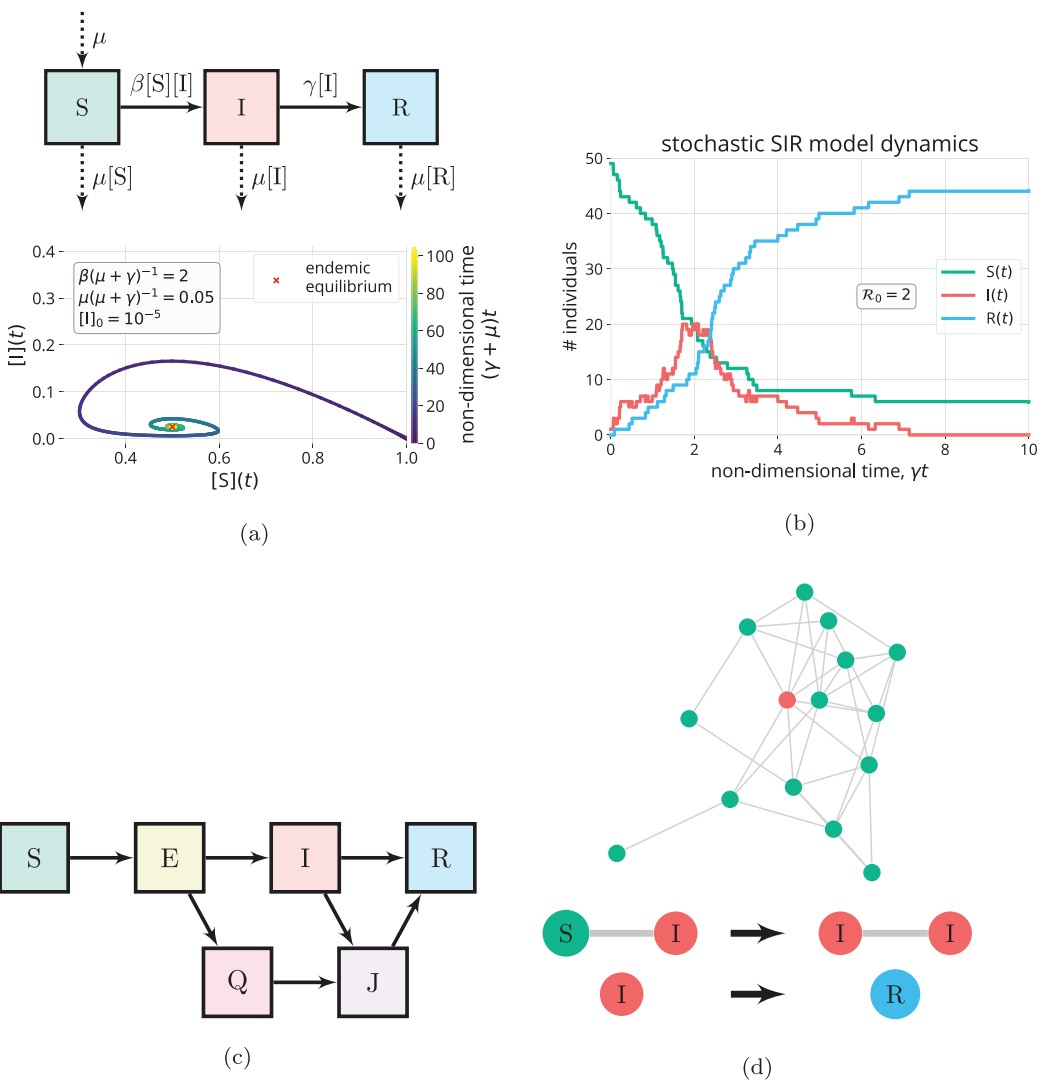

**Figure 6 Extensions to the SIR model.** (A) Births and natural deaths (balanced, with rate constant μ) are introduced to the SIR model through the flows in/out of the compartments denoted by the dashed arrows (top). With births and deaths, the trajectory of the solution in the phase plane (bottom) shows $[I](t)$ can exhibit damped oscillations that settle on an endemic equilibrium ($\mu(\mu + \gamma)^{-1}$ unrealistically large to clearly see $\lim_{t\to\infty}[I](t) > 0$) (*Hethcote, 2000*). (B) A simulation of the (stochastic) SIR continuous-time Markov chain model (*Allen, 2017*) for a small population of 50 individuals, with one initially infectious. The state variables S, I and R are discrete counts of individuals in the compartments. (C) Introducing additional compartments. For example, exposed individuals have been exposed to and infected by the disease, but cannot yet transmit the disease, due to a latent period. Owing to control measures, quarantined (Q) and isolated (J) individuals make contacts with a reduced frequency. (D) Contact network models. (top) Nodes represent individuals, and each is in the S, I, or R state. Edges, along which disease transmission can occur, represent contacts. (bottom) The state changes of nodes in an SIR contact network model (*Kiss, Miller & Simon, 2017*).

infected patients (*Bjørnstad et al., 2020a*). Evolution of the pathogen could also cause β and γ to change with time (*Lion & Metz, 2018*). Reducing the transmission rate constant β and/or the average time period of infectiousness $\gamma^{-1}$ reduces the replacement number $r$ (see Eq. (4)); if $r$ is reduced below one, the prevalence of infectious folks will decrease.

Analogously, in chemical kinetics, time-varying reaction rate constants arise when the temperature in the reactor changes with time (*Fogler, 2010*).

### Stochasticity

As opposed to the deterministic differential Eqs. (1)–(3), we can introduce randomness into the SIR model to account for the stochastic and uncertain nature of human interaction and disease transmission (*Allen, 2008*, *2017*; *Britton, 2010*). Stochastic epidemic models aim to describe the probabilistic distribution of outcomes, for example, the distributions of $[R]_\infty$ and $\max_t [I](t)$ (*Allen, 2008*; *Britton, 2010*). Stochasticity can be particularly important to model for small populations and in the early stage of an epidemic, when there are small numbers of infectious individuals (*Britton, 2010*). Analogously, when modeling chemical reaction dynamics, if the reactants are not abundant, such as in a biological cell, discreteness and stochasticity can be important to model (*Erban, Chapman & Maini, 2007*; *Gillespie, Hellander & Petzold, 2013*). See Fig. 6B for an example of a stochastic simulation of the SIR model.

### More realistic probabilistic distributions of the infectious time period

The probabilistic implication of first-order decay in Eq. (3) for a single infectious individual is that their time period of infectiousness is an exponentially distributed random variable with mean $\gamma^{-1}$ (*Brauer, 2008*). More realistic probabilistic distributions of the time period of infectiousness can be built into the SIR model (*Hethcote & Tudor, 1980*; *Lloyd, 2001*). In the realm of biochemical kinetics, such non-exponential waiting times could arise, for example, between gene transcription events, owing to several steps involving transcription factors, cofactors, chromatin remodeling, etc. (*Pedraza & Paulsson, 2008*; *Haberle & Stark, 2018*).

### Additional compartments

We can introduce additional compartments and further divide the population to (i) account for heterogeneity in the population, (ii) model public health interventions, and (iii) include other characteristics of the disease. Members of different compartments can then have different contact patterns, transmissibilities, and/or recovery times. For example, the SEQIJR model (*Gumel et al., 2004*) introduces three additional compartments: Exposed (E), Quarantined (Q), and Isolated (J). Usually, the E compartment is included to model the latent period of a disease; exposed individuals have been exposed to and infected by the disease but are not yet infectious (*Brauer, 2008*; *Martcheva, 2015b*). The Q and J compartments are included to model the control measures of quarantining exposed individuals and isolating infectious individuals, respectively. Members of the Q and J compartments contact other members of the population with reduced frequencies, compared to those in the E and I compartments, respectively (*Brauer, 2008*). See Fig. 6C. As another example, we can partition the infectious compartment into two distinct categories of infectious folks: symptomatic and asymptomatic (*Vivas-Barber, Castillo-Chavez & Barany, 2014*), which can have different frequencies of contacts, transmissibilities, and recovery rates. Finally, to account for different mixing patterns among different age groups (*Brauer, 2008*), age-structured compartmental models

partition the S and I compartments into age cohorts. Notably, introducing additional compartments to the SIR model is much like introducing additional reactive chemical species into a chemical kinetics model.

Additional compartments complicate the derivation of the basic reproduction number $\mathcal{R}_0$ from the model (*Van den Driessche, 2017*). Still, the peak prevalence of infectious folks and final size of the epidemic increase with $\mathcal{R}_0$ in most models (*Van den Driessche & Watmough, 2008*). Notably, a more precise definition of $\mathcal{R}_0$ is needed if compartments are introduced to account for heterogeneity in the population (*Diekmann, Heesterbeek & Metz, 1990*).

### Spatial heterogeneity

We can model spatial heterogeneity of an epidemic, that is, spatially-dependent [S], [I] and [R], by treating space as discrete (*Van den Driessche, 2008*) or continuous (*Martcheva, 2015a*). Modeling the spatial movement of susceptible and infectious individuals in a continuous space as a diffusive process results in reaction-diffusion equations (*Martcheva, 2015a*), also commonly used in the chemical sciences. Compartmental, epidemic models of metapopulations allow travel between spatially segregated regions and resemble models of multiple reactors connected with pipes that permit flow between them (*Van den Driessche, 2008*; *Arino et al., 2005*; *Balcan et al., 2010*).

### More detail/structure in the contact patterns

Network epidemiological models allow us to study how structure and heterogeneity in the contact patterns within a population impact the dynamics of an epidemic (*Kiss, Miller & Simon, 2017*; *Bansal, Grenfell & Meyers, 2007*; *Keeling & Eames, 2005*; *Newman, 2002*). In a typical contact network model, each individual in a population is represented by a node, and each interaction/contact between a pair of individuals is represented by an edge. The state of an SIR network at a given time is specified by the disease status (S, I, or R) of each node. The disease can be transmitted along an edge if one of the nodes is infectious and the other is susceptible (*Kiss, Miller & Simon, 2017*) (See Fig. 6D).

The dynamics of disease transmission on a contact network depend significantly on the structure of the network, for example, on its degree distribution, node clustering, and correlations between the degrees of connected nodes (*Bansal, Grenfell & Meyers, 2007*). Intuitively, the dynamics of an SIR model on large, static, random $k$-regular contact networks (a random network where each node has $k$ neighbors) closely approximate those of Eqs. (1)–(3), which assume homogeneous mixing (*Bansal, Grenfell & Meyers, 2007*). In contrast, the dynamics of SIR models on highly heterogeneous contact networks, such as those with heavy-tailed degree distributions that capture "superspreaders" (*Lloyd-Smith et al., 2005*; *Meyers et al., 2005*), deviate significantly from Eqs. (1)–(3).

Interestingly, directed networks can model asymmetric contacts that can only transmit the disease one-way, such as donated blood transfusions (*Keeling & Eames, 2005*; *Allard et al., 2020*).

A contact network model such as in Fig. 6D only implicitly accounts for spatial heterogeneity. To more explicitly account for spatial heterogeneity, contact networks can

be encoded as a dynamic, bipartite graph with two classes of nodes: individuals and locations. Edges then connect individuals with locations, and disease transmission can occur between people that are in the same location. (*Eubank et al., 2004*) Other agent-based models are fully spatially explicit and track the location of individuals as they move between households, schools, and workplaces (*Ferguson et al., 2005*). Notably, to construct such contact network models, very detailed data is required (*Ajelli et al., 2010*).

### Vectors that transmit the pathogen from host to host

Some infectious diseases are primarily transmitted from one host to another host by living *vectors* that can acquire and carry the infectious agent. For example, mosquitoes can acquire an infectious agent (e.g., the virus that causes dengue fever or the parasite that causes malaria) from feeding on the blood of an infected human, then transmit the infectious agent to another, susceptible human when feeding on their blood. SIR-like models of diseases transmitted by vectors include an incidence term $\beta \, [S][I_v]$, where $[I_v]$ is the concentration of infectious vectors and $\beta$ includes the frequency that the vector bites hosts and the probability of transmission conditioned upon a bite. (*Martcheva, 2015c*, *Smith et al., 2012*).

## CONCLUSIONS

Mathematical models of the dynamics of disease transmission are used to forecast epidemics and assess mitigation strategies. We provided an exposition on the classic SIR dynamic model of disease transmission and highlighted the analogy between disease transmission and an autocatalytic reaction with catalyst deactivation. This analogy links together the fields of chemical kinetic and epidemic modeling to enable knowledge exchange between the two research communities. Moreover, the analogy could be the basis for an engaging, experience-based approach to learning chemical kinetics in the classroom (*Sucre-Rosales et al., 2020*) and illustrate how concepts in one field can transfer, aid understanding, and generate insights to/in another field.

## ACKNOWLEDGEMENTS

Thanks to Prof. Fred Brauer for very helpful comments on the preprint.

### Funding

This work was supported by the National Science Foundation (Award #1920945). The funders had no role in study design, data collection and analysis, decision to publish, or preparation of the manuscript.

### Grant Disclosures

The following grant information was disclosed by the authors:
National Science Foundation: #1920945.

## Competing Interests

The authors declare that they have no competing interests.

## Author Contributions

- Cory M. Simon conceived and designed the experiments, performed the experiments, analyzed the data, performed the computation work, prepared figures and/or tables, authored or reviewed drafts of the paper, and approved the final draft.

## Data Availability

The Julia codes to reproduce all plots in this article are openly available on GitHub at: https://www.github.com/SimonEnsemble/sir_model.

## Supplemental Information

Supplemental information for this article can be found online at http://dx.doi.org/10.7717/peerj-pchem.14#supplemental-information.

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
