# Peer review of "The SIR dynamic model of infectious disease transmission and its analogy with chemical kinetics"

_PeerJ Physical Chemistry, doi:10.7717/peerj-pchem.14_

## Round 0.1 · original submission · Minor Revisions

Please particularly take into account the comments of reviewer 1 regarding the discussion of chemical analogies to extended SIR models. Also, all subsection titles should be capitalized.

Reviewer 1 ·

Basic reporting

no comment

Experimental design

The manuscript examines the analogy between differential equations and their solutions in chemical kinetics and epidemiological models. This goal is accomplished satisfactorily. The discussion could be further improved by giving engineering/chemical kinetics analogues to the extensions of the SIR model, if such analogues exist.
Perhaps quarantine in the SEQIJR model could be related to the presence of an absorber material in a reactor?
I am unsure how time-varying parameters and births, death, and loss of immunity could be related to the chemical kinetics/engineering, but perhaps the author has an idea.

Validity of the findings

no comment

Additional comments

I found the analogy between kinetic and epidemiological models very unexpected, but also very interesting. The manuscript presents this analogy in a very instructive way and I think it is a valuable read for anyone involved in physical chemistry in some way.

Reviewer 2 ·

Basic reporting

This paper offers a pedagogical introduction to the Susceptible-Infectious-Recovered (SIR) epidemic model for readers familiar with chemical kinetic models. By drawing analogies between chemical kinetic and compartmental epidemic models, the Author aims to foster the exchange of ideas between disparate research communities.

The paper is well-organized, clearly written, and technically accurate. I would be happy to share a published version with my colleagues to use in lectures for senior undergraduate chemical engineering students. Moreover, the highlighted extensions of the simple SIR model may serve to stimulate research professionals in the area of chemical kinetics to seek opportunities to apply their expertise in “new” domains.

Experimental design

This paper is a review of prior work.

Validity of the findings

The presentation of the SIR model is clear and accurate.

---

## Round 0.2 · accepted · Accept

The revisions have adequately addressed the concerns of the reviewers and overall improved the manuscript further.